# Improving Photostability of Photosystem I-Based Nanodevice by Plasmonic Interactions with Planar Silver Nanostructures

**DOI:** 10.3390/ijms23062976

**Published:** 2022-03-10

**Authors:** Marcin Szalkowski, Dorota Kowalska, Julian David Janna Olmos, Joanna Kargul, Sebastian Maćkowski

**Affiliations:** 1Institute of Physics, Faculty of Physics, Astronomy and Informatics, Nicolaus Copernicus University in Toruń, ul. Grudziądzka 5, 87-100 Toruń, Poland; m.szalkowski@intibs.pl (M.S.); dorota@fizyka.umk.pl (D.K.); 2Institute of Low Temperature and Structure Research, Polish Academy of Sciences, ul Okólna 2, 50-422 Wrocław, Poland; 3Solar Fuels Laboratory, Centre of New Technologies, University of Warsaw, ul. Banacha 2C, 02-097 Warsaw, Poland; julian.janna.olmos@uj.edu.pl (J.D.J.O.); j.kargul@cent.uw.edu.pl (J.K.)

**Keywords:** photosystem I, plasmonic interactions, silver island film, biophotoelectrodes, photostability

## Abstract

One of the crucial challenges for science is the development of alternative pollution-free and renewable energy sources. One of the most promising inexhaustible sources of energy is solar energy, and in this field, solar fuel cells employing naturally evolved solar energy converting biocomplexes—photosynthetic reaction centers, such as photosystem I—are of growing interest due to their highly efficient photo-powered operation, resulting in the production of chemical potential, enabling synthesis of simple fuels. However, application of the biomolecules in such a context is strongly limited by the progressing photobleaching thereof during illumination. In the current work, we investigated the excitation wavelength dependence of the photosystem I photodamage dynamics. Moreover, we aimed to correlate the PSI–LHCI photostability dependence on the excitation wavelength with significant (ca. 50-fold) plasmonic enhancement of fluorescence due to the utilization of planar metallic nanostructure as a substrate. Finally, we present a rational approach for the significant improvement in the photostability of PSI in anoxic conditions. We find that photobleaching rates for 5 min long blue excitation are reduced from nearly 100% to 20% and 70% for substrates of bare glass and plasmonically active substrate, respectively. Our results pave promising ways for optimization of the biomimetic solar fuel cells due to synergy of the plasmon-induced absorption enhancement together with improved photostability of the molecular machinery of the solar-to-fuel conversion.

## 1. Introduction

Continuous growth of global energy consumption, accompanied with fossil fuel resources depletion and the need to limit the negative outcomes of the combustion thereof (CO_2_ emissions and air pollution), has been motivating the development of alternative energy sources [1]. In addition to nuclear or geothermal energy solutions, renewable sources are considered important contributors to the future energy mix. In particular, solar energy is highly promising and has still not been fully exploited as an energy source, taking into account the huge amount of energy, which is delivered to the surface of the Earth (60–250 W m^−2^, despite significant variation with latitude, and with cyclic annual and daily changes) [2,3]. Although the most widespread and commercially utilized solar cells are based on semiconductors (predominantly silicon, with recent emerging of perovskite-based solar cells [4,5]), their operation is limited to the solar-induced separation of electric charges and, therefore, production of electric energy. In recent years it has also been proposed to develop energy sources inspired by photosynthesis, the process optimized by nature over billions of years of evolution to harvest solar energy and convert it to a usable form of chemical energy [6]. The working horses of the photosynthetic apparatus are photosystem I (PSI) complexes—sophisticated pigment–protein complexes arranged with high precision in order to obtain functional systems capable of performing the first steps of the solar-to-chemical energy conversion. These include harvesting light as well as charge separation and production of electrochemical potential on the two sides of the photosynthetic membrane [7,8,9,10]. The most important reasons underlying this approach, include (1) extremely high quantum efficiency of the energy conversion (close to 100%, i.e., nearly each photon absorbed by PSI results in charge separation); (2) operation not limited to the production of photoinduced voltage—similarly as in the photosynthetic apparatus, bioinspired catalytic systems may produce solar fuels and chemicals, such as hydrogen, formate, carbon monoxide, or alcohols [11,12,13,14,15]. These products are not only easier to store and transport than electric energy, but such a functionality may also improve CO_2_ neutral final balance.

In recent years, much attention has been devoted to exploring the potential of PSI complexes in the context of energy conversion, and several designs of the PSI-based photoelectrodes or devices have been proposed, designed, characterized, and validated, demonstrating photocurrent densities in the range from tens of nA cm^−2^ up to around 1 mA cm^−2^ [16,17,18,19,20,21,22,23,24]. In particular, it was suggested to improve the function of the PSI using plasmon excitations induced in metallic nanoparticles [25,26,27,28]. Localized surface plasmon resonance (LSPR) is formed by collective oscillations of the plasma gas in the nanoscale-sized particles, typically of noble metals, and provides a way to modify the optical properties of the nearby objects [29,30]. It was reported that large PSI fluorescence enhancements (even by a factor of 200) are possible when these complexes interact with plasmonically active metallic substrate [31]. Furthermore, not only rates of absorption or emission can be modified due to such interaction. It has been reported recently that stimulation of the oriented multilayer of PSI complexes assembled on graphene with plasmon excitations induced in the silver islands leads to significant enhancement of the photochemical performance of such systems, including higher photocurrents (up to 1.5 µA cm^−2^ of photocurrent density was reported, giving fivefold improvement versus non-plasmonic reference) [32].

Despite substantial advantages of utilizing photosystems in bioinspired solar fuel cells, there are also limitations inherently connected with the design of an efficient and economically reasonable device. One of the essential issues that needs to be addressed is the durability thereof, in particular, the photostability. It is well-known that all biomolecular emitters exhibit photobleaching resulting in the gradual irreversible photodegradation and thus the loss of the function [33,34]. Addressing this issue requires understanding of the processes responsible for the photoinduced damage of the molecules. In addition, the temperature increase caused by solar radiation, which results in protein denaturation [35,36,37], is another factor, which limits the photostability of biomolecules, together with interactions with the reactive oxygen species (ROS). Two of them, i.e., singlet oxygen (^1^O_2_) and superoxide radical (O_2_^−^), are produced due to reaction of oxygen molecules (^3^O_2_) with biomolecules in triplet excited state (in the processes of energy transfer and electron transfer, respectively) [38,39]. Interaction of biomolecules with ROS results in irreversible damage and, thus, loss of their functionality [40,41]. However, photoinduced production of ROS by the dyes is highly desired in the context of application of such mechanisms in photodynamic therapy, where controlled intentional local production of ROS is utilized to annihilate cancer cells [42,43].

In this work, we investigated the photostability of the extremophilic PSI–LHCI complex isolated from a volcanic red microalga *Cyanidioschyzon merolae* as a function of illumination wavelengths corresponding to the solar spectrum. We show that the excitation wavelength influences not only the PSI emission intensity, but also the photodamage rate of the photosystems. Furthermore, we analyzed the impact of the plasmonically active substrate, i.e., silver island films (SIFs), as well as the presence of oxygen on the photostability of the PSI–LHCI complex. Our study identifies two competing effects related to utilization of the SIF substrates: (1) PSI–LHCI fluorescence enhancement and (2) increased PSI photobleaching rate. Our analysis of timetraces of PSI–LHCI emission intensity demonstrates that benefits from applying plasmonically active SIF substrates (with the fluorescence intensity enhancement reaching factors of around 50) surpass any negative effects resulting from intensified PSI–LHCI photodamage. This observation is highly relevant for the biosolar applications of this complex. Finally, we demonstrate the significant (up to fivefold) improvement in the PSI–LHCI photostability in anoxic (vacuum) conditions, pointing towards the crucial role of the highly reactive singlet oxygen species in inducing the photodamage of the PSI–LHCI complex. These results should be considered when designing efficient and stable bioinspired solar cells.

## 2. Results and Discussion

We investigated eukaryotic photosystem I–light harvesting complex I (PSI–LHCI) isolated from a red microalga *C. merolae*. This photoactive protein–pigment complex, in contrast to its cyanobacterial counterpart, forms monomers in the photosynthetic membrane, in which the core domain coordinating all the electron transport cofactors is associated with an asymmetrically bound LHCI system composed of 3–8 Lhcr subunits [44,45,46]. Each red algal PSI–LHCI monomer is composed of 158–182 Chl *a* molecules, forming the light harvesting system, as well as the charge separation chain and P700 special pair [44,45,46,47]. In addition to Chl *a* fluorophores, the red algal PSI–LHCI complex contains 46 carotenoids (Cars) including 21 β-carotenes and 25 zeaxanthins [45,46]. Each Lhcr subunit of the LHCI system contains 11–13 Chl *a* and 5 zeaxanthin molecules. The specific pigment molecules identified in the recent X-ray and cryo-EM structures of the *C. merolae* PSI–LHCI complex are proposed to form discrete energy transfer pathways differing to those in the higher plant complex [45,46] and varying with light intensity [48]. Recently, the photoprotective mechanism has been identified in the red algal PSI–LHCI complex based on the accumulation of low-energy (red) Chl *a* molecules, mainly in the LHCI subunits under light-limiting conditions that exert a dual function: as the light harvesting molecules in low light or as energy traps under high light conditions [48]. Moreover, in high light, the LHCI antenna appears functionally decoupled from the PSI reaction center, as shown by the recent time-resolved spectroscopic studies on isolated PSI complexes [48,49].

The aim of this study was: (1) to investigate the effect of excitation wavelength on the photostability of red algal PSI–LHCI immobilized on the glass substrate, (2) correlate the PSI–LHCI stability with plasmonic enhancement of fluorescence (or absorption) of this complex at various excitation wavelengths, and (3) to determine the impact of oxygen presence on the photostability of PSI–LHCI deposited on both substrates. We hypothesize that plasmonic enhancement of optical functionality of PSI–LHCI may lead to improved photostability. 

### 2.1. Spectral Characterization of PSI–LHCI and SIF

Firstly, we examined the optical properties of red algal PSI–LHCI in solution and its spectral overlap with the plasmonic substrate of silver island film (SIF). This metallic nanostructure is composed of the silver islands irregular in shape, with sizes varying in the range of 50–250 nm, which are densely deposited on a glass substrate [50].

Figure 1 shows the absorption spectrum of the red algal PSI–LHCI complex (black line), in which contributions of the two types of pigments can be identified: peaks corresponding to the absorption of Chl *a* molecules (with Soret band in violet and blue spectral ranges, and Q_y_ band in the red part of the visible range) and the Cars (absorbing mainly yellow light). On the other hand, emission spectrum of the aqueous PSI–LHCI solution (excited at 405 nm), shown in red, exhibits strong peaks with a maximum around 680 nm, with characteristic broadening towards longer wavelengths.

In Figure 1, both PSI–LHCI spectra are confronted with the SIF extinction spectrum (blue line), which exhibits the plasmon resonance band with maximum around 430 nm (overlapping well with the Soret absorption band of the Chl *a* molecules); however, its tail extends toward longer wavelengths covering the whole visible spectral range. In particular, emissions of PSI–LHCI are located in the spectral range where interaction with plasmon excitations in SIF can be expected. Therefore, both absorption and emission rates of PSI–LHCI may be influenced by coupling with plasmon excitations in SIF.

### 2.2. Excitation Wavelength Dependence

In order to determine the photostability of PSI–LHCI complexes on a glass substrate we continuously illuminated selected spot on the sample (previously not excited) with focused laser light and measured subsequent emission spectra (each of them collected per 1 s) for 300 s. This procedure was repeated for each of 5 excitation wavelengths corresponding to various bands of the PSI–LHCI absorption (405 nm, 485 nm, 570 nm, 580 nm, and 610 nm). In each case such a routine, repeated for 10 various spots on the sample surface, was applied in order to provide statistically relevant set of data.

Representative PSI–LHCI spectra measured for three excitation wavelengths are shown in Figure 2 (panels a, b, and c for excitation at 485 nm, 580 nm, and 610 nm, respectively). All the plots show the spectra measured at the beginning of the illumination (0 s, red line), as well as the spectra measured on the same spot after 10 s, 30 s, 60 s, 120 s, 180 s, and 300 s from the start. Intensity of the initial spectra vary with the excitation wavelength, as one can expect from the relation between the laser lines and PSI–LHCI absorption spectrum shape. Moreover, the rate of emission photobleaching varies significantly with the excitation wavelength. Clearly, the slowest photobleaching is observed for 580 nm excitation, whereby the laser light is poorly absorbed by these complexes resulting in the weakest emission. A much faster rate of photobleaching was observed for the other excitation wavelengths. Such a variation in the PSI–LHCI photostability, inversely proportional to the PSI–LHCI absorption efficiency at given wavelength rather than with the energy of the exiting photons, suggests that only light absorbed by the PSI–LHCI complexes results in their photobleaching. Thus, the observed photodamage is associated with the photobleaching of the excited PSI–LHCI particles rather than with the thermal degradation caused by local heating of the sample by the laser radiation.

In order to illustrate the full temporal evolution of the PSI–LHCI emission intensity during the experiment, we integrated the intensity of the spectra collected in each of the measured time frames (from 650 to 800 nm). The results are shown in panels (d–f) in Figure 2. The logarithmic scale was used for the vertical axis. The integrated emission kinetics emphasize the universal property of the PSI–LHCI photobleaching dynamics—the rate of the photodegradation of the photoactive complexes is not constant in time. For all excitation wavelengths the fastest rate was observed at the beginning of the experiment, followed by much slower (albeit still progressing) photobleaching. 

### 2.3. The Influence of the SIF Substrate

In order to investigate the impact of the SIF substrate on the photostability of the PSI–LHCI complexes, we prepared analogous samples, but with a layer of PSI–LHCI spin-casted onto the SIF-covered glass. The methodology of the experiment was identical with the one applied for PSI–LHCI on glass. Figure 3 compares the results obtained for 580 nm excitation for two substrates, glass, and SIF (panels a and b, respectively). One of the conspicuous impacts of the SIF interaction with the red algal PSI–LHCI is significant (around 50-fold) PSI–LHCI fluorescence enhancement compared with the sample of PSI–LHCI on bare glass. Moreover, spectra maxima are slightly blue-shifted, as reported previously for similar structures [27,31], which may reflect minute structural changes in PSI upon its interaction with SIF. Such an enhancement of the emission intensity can be attributed to the interaction of the photoactive molecules with plasmon excitations in SIF, with the dominant role of the plasmon-induced gaining of the PSI–LHCI absorption rate, as discussed previously [31,32]. Furthermore, the photodegradation, especially at the early moments of the illumination, is occurring much faster for the photosystems interacting with plasmon interactions than for the reference sample prepared on the bare coverslip—after 10 s long illumination, only ~30% of the initial number of PSI–LHCI complexes is still emitting. Faster photobleaching of samples prepared on the SIF also may be attributed to the plasmon-enhanced absorption of PSI–LHCI complexes located in the volume around the silver islands, where strongly enhanced electric fields occur and, thus, intensified interactions of the photosystems with light should be expected. Increased probability of PSI–LHCI complexes to be promoted to the excited state leads in turn to corresponding increase in the photobleaching probability.

### 2.4. The Impact of the Oxygen Presence on the PSI–LHCI Photostability

Results described so far were collected for the samples kept in ambient conditions (i.e., in the presence of oxygen). In the next step, we performed a similar set of the measurements for the samples enclosed in a vacuum chamber, with the pressure lowered to 1.4 × 10^−8^ bar. Results obtained for PSI–LHCI complexes on glass and SIF substrate, are presented in the Figure 3 (panels c and d, respectively). In this case, we find significant fluorescence enhancement of the PSI–LHCI interacting with SIF; although, slightly lower emission intensity was obtained for both sample types in vacuum conditions compared to the results obtained in ambient conditions. Interestingly, dramatic changes in the PSI–LHCI photobleaching dynamics can be observed. In the case of the photosystems deposited on glass photobleaching of the PSI–LHCI emission is negligible, and the fluorescence intensity only slightly decreases during the laser illumination. This is, however, not the case of the sample prepared on SIF substrate, where photobleaching still occurs in a vacuum, but its rate is significantly reduced. Indeed, even 300 s long laser light irradiance leads to lower photodegradation when PSI–LHCI is kept in the vacuum than observed after 10 s of illumination for the same sample in ambient conditions.

In order to compare the influence of the SIF substrate and the presence of oxygen on the photostability of the PSI–LHCI complexes we combined the curves illustrating temporal changes in the integrated emission intensity of consecutive spectra measured during the experiment. The results of the measurements carried out using excitation at 485 nm, 580 nm, or 610 nm are shown in Figure 4 (panels a–c). General observations according to (1) plasmonic enhancement of PSI–LHCI emission intensity, (2) faster photodegradation of photosystems deposited on the SIF than on glass, and (3) significant reduction in the photobleaching for the PSI–LHCI placed in the vacuum are all still valid for all excitation wavelengths. The photobleaching dynamics varies, however, with the excitation wavelength. First of all, the excitation at 485 nm or 610 nm exerts in the most destructive impact on the PSI–LHCI, regardless of the conditions (see Figure 4). Conversely, the most stable PSI–LHCI emission (although also the weakest) was found for the excitation at 580 nm. These results support the hypothesis that only light absorbed by the PSI–LHCI complex (and thus inducing the excited state of the P700 reaction center of PSI) is involved in the photobleaching. Other possible effects such as local sample heating by scattered laser radiation seem to have less importance for the observed photobleaching kinetics. The PSI–LHCI photostability improvement observed in vacuum conditions is particularly impressive for the samples prepared on glass substrates, where nearly total elimination of the photodegradation was found for all of the excitation wavelengths. This, in turn, suggests a crucial role of the oxygen on the photostability of PSI–LHCI.

A similar effect of varying oxygen concentrations on the stability of PSI immobilized on the gold electrode was observed by Zhao and colleagues [51] who showed that elimination of reactive oxygen species leads to improvement in the long-term stability of PSI-based biophotocathode [51]. In high light, ^1^Chl* species often undergo intersystem crossing to form the Chl triplet state (^3^Chl*). The Chl triplet reacts with molecular oxygen to form highly reactive oxygen species, including singlet oxygen (^1^O_2_), when excitations are not rapidly quenched. This can occur when the capacity for productive photochemical quenching is exceeded and non-photochemical quenching is insufficient or not yet sufficiently turned on [52]. Singlet oxygen, due to extremely high reactivity, is the source of degradation of the nearby proteins, lipids, and emitters [52,53].

The dramatic improvement of the PSI–LHCI photostability in the vacuum observed in our study points toward singlet oxygen production as the dominant mechanism responsible for the observed photobleaching. However, even despite the absence of the oxygen there can be observed very slow photodegradation, which can be attributed to the interaction of PSI–LHCI complexes with a small number of the oxygen molecules still present in the chamber or locked in the polymer matrix during the sample preparation, as well as to the local laser-induced heating, which may result in the thermal irreversible denaturation of the proteins [35,54]. Nonetheless, the impact of these effects on the photostability of PSI–LHCI complexes is negligible as compared to photobleaching due to the influence of oxygen. In the case of photosystems deposited on the SIF excited with 580 nm and 610 nm a minute, a gradual decrease in the PSI–LHCI emission intensity can be observed in the vacuum. Less improvement was found for 485 nm excitation, where, especially during the first 30 s of illumination, photobleaching clearly occurs. Afterwards the emission becomes rather stable. It, also, as already concluded for samples prepared on glass, points out the essential role of oxygen’s presence on the photostability of photosystems, and the faster photobleaching observed for SIF substrates is associated with an increased absorption rate. As a result, PSI–LHCI complexes absorb more radiation, and therefore they more often stay statistically in the excited state compared to PSI–LHCI on glass. More absorption events and, thus, longer time spent in the excited state results in an increased probability of the interaction with molecular oxygen, which leads to an enhanced production of destructive singlet oxygen species and stronger PSI–LHCI photobleaching. Moreover, the photobleaching is considerably faster at the beginning of the experiment and then its dynamics slows down, reaching a plateau in some cases. This effect might be related to the sample architecture, where a layer of PSI–LHCI complexes is formed in a polymer matrix. Then, at the beginning of illumination the photobleaching of the outer PSI–LHCI complexes, the ones exposed to free oxygen molecules, dominates. Photosystems placed in deeper layers of the polymer matrix are preserved better against the oxygen, what results in saturated photostability afterwards. However, the samples prepared on SIF, even in an oxygen-free environment, still exhibit some photobleaching compared to the PSI on glass. Here, also, it can be attributed to the interaction of the PSI–LHCI with oxygen molecules in the polymer matrix and to the thermal-induced photodamage. The latter seems to be more probable in this case, as metallic nanoparticles excited with wavelengths corresponding to their plasmon resonance band dissipate energy as heat, leading to local temperature increase within the laser illuminated volume, and thus enhancing thermal denaturation of the proteins [35,54].

Other meaningful information about the dynamics of the two dominant effects involved—PSI–LHCI plasmonic enhancement and the photobleaching of these complexes—can be obtained from the temporal changes in the PSI–LHCI fluorescence intensity ratios, as plotted in Figure 4 (panels d–f). The curves were defined as the ratio of the integrated intensities obtained at the given moments in time since the beginning of the illumination for the samples prepared on the SIF and prepared on the glass. Such dependencies were determined for the samples measured both in ambient conditions, as well as in the vacuum. We find that there is relatively fast decrease in the intensity ratio in the first tens of seconds, and then the ratio stabilizes. Importantly, much higher intensity ratios were found for the samples in the vacuum. In the case of 580 nm and 610 nm excitations, they approach stable value of around 50, while in the case of blue excitation the plateau is established at much lower levels—around 30. For this wavelength there is also the largest disproportion between intensity ratios recorded in vacuum and in ambient conditions. Such temporal intensity ratios are the measure of relative changes in the photobleaching of PSI–LHCI with and without plasmonic interaction involved. It can be concluded that at first moments of the irradiation the photodegradation of PSI–LHCI complexes are more effective on SIF; however, as the experiment progresses, the dynamic of the photobleaching approaches the similar rate, irrespective of the substrate. This conclusion is valid for both sample environments. 

### 2.5. Statistical Analysis

In order to quantitatively compare the impact of the various conditions on the photobleaching of photosystems, we calculated the percentage decrease in the integrated PSI–LHCI intensity, *P*, as defined:(1)$$P=\frac{I_{0\;s}-I_{300\;s}}{I_{0\;s}}\times 100\%,$$
where $I_{0\;s}$ is the integrated PSI–LHCI emission intensity measured at the beginning of the illumination and the $I_{300\;s}$ is, analogously, the value measured after 300 s of the illumination. In other words, *P* provides information about the relative decrease in the emission intensity during the experiment, and the boundary conditions, *P* = 0% and *P* = 100%, correspond to no photobleaching and total photodamage, respectively. We analyzed results obtained for all of the wavelengths used in the experiments, and the procedure was performed for all of the measured spots on the samples. Excitation wavelength dependencies of photobleaching factors, *P*, are presented in Figure 5a. There is an emerging clear trend in the PSI–LHCI photodegradation efficiency. The most pronounced photobleaching was found for the SIF sample measured in ambient conditions, for which nearly 100% photodamage occurred after 300 s of irradiation, independent of the excitation wavelength. In the case of 405 nm and 485 nm illuminations best fitted into the Soret absorption band of the Chl *a*, the photobleaching of photosystems deposited on glass is similar to the SIF sample; however, for other excitation wavelengths (especially 570 nm and 580 nm, corresponding to the region of low PSI–LHCI absorption) photodamage is less efficient for the samples on glass. On the other hand, providing oxygen-free conditions yields dramatic improvement in the photostability, including the SIF sample, which performs even better than the sample on glass in ambient conditions. A similar trend was also found for excitations at 570 nm and 580 nm. The best photostability was observed for samples on glass placed in the vacuum, for which a photobleaching factor below 50% (~20% for most of the wavelengths used) was observed.

Regarding general observations from the experiments carried out for the PSI–LHCI samples, we identify two effects with an opposite impact on the emission intensity of the biomolecules – fluorescence intensity is strongly enhanced due to plasmonic interactions, however, with a simultaneous faster decrease in emission intensity due to more efficient photobleaching. To determine which of these dominates, we compared total intensity emitted by the photosystems during the whole 300 s long experiments. The average results for all of the analyzed variants are shown in Figure 5b. The presented data confirm higher total intensities obtained for SIF-based samples than for their glass analogues with total intensities measured for both glass samples being far below those corresponding to the SIF structures. It leads to the conclusion that the benefits of plasmonically active substrates (i.e., significant, ~50 times enhancement of fluorescence intensity) dominate over any negative impact caused by faster photobleaching. Indeed, even over 10 times more photons were emitted by the PSI–LHCI complexes deposited on SIF in vacuum conditions versus the reference on glass.

These results suggest that incorporation of plasmonically active nanostructures yield gains in absorption rates of biomolecules that are highly beneficial from the point of view of designing solar cells or solar fuel cells, and should translate into more efficient solar energy conversion. Even in ambient conditions, the profits resulting from plasmonic interactions overcome drawbacks associated with reduced photostability. Clearly, such a balance can be significantly and positively shifted when vacuum conditions are applied and the most destructive factor—oxygen—is eliminated. Therefore, the presented results are of high importance from the point of view of the overall efficiency of the designed biomimetic solar fuel cells in a scalable way by plasmon-induced large enhancement of the absorption rate of the PSI–LHCI molecules, which is expected to result in an increased number of photon-to-separated charges events and, thus, gained photocurrents. Furthermore, such enhancement still may be amplified by providing highly controlled spatial organization of the components, for example utilizing silver substrate functionalization. The presented results also point toward the necessity of designing such solar cells in a way that provides an anoxic environment for the reaction centers carrying out the initial steps of the energy conversion in order to vitally limit the gradual photobleaching thereof.

## 3. Materials and Methods

### 3.1. PSI–LHCI Isolation and Purification

The procedure of *C. merolae* culture growth, thylakoid isolation, solubilization, and PSI–LHCI complex purification was performed, as described in [44], using anion-exchange chromatography. Briefly, the crude PSI–LHCI fraction was eluted from the DEAE TOYOPEARL 650 M column, loaded with solubilized thylakoids, with 0.09 M NaCl, then applied onto the DEAE TOYOPEARL 650S column. The pure PSI–LHCI fraction was eluted with a continuous 0–0.2 M NaCl gradient in the carrier buffer, as described in [44]. The PSI–LHCI sample was concentrated to 1 mg/mL Chl*a* and further purified on the desalting Superdex G-25 column in Buffer B (40 mM HEPES-NaOH, pH 8, 3 mM CaCl_2_, 25% (*w*/*v*) glycerol, 0.03% (*w*/*v*) DDM), followed by an anion exchange chromatography step using a UNO™ Q12 column, following the procedure described in [44]. The fractions containing the pure PSI–LHCI supercomplex were collected and concentrated to 2–5 mg/mL Chl*a*, snap-frozen in liquid N_2_ and stored at −80 °C prior to use.

### 3.2. Samples Preparation

Bare non-conductive glass coverslips were carefully cleaned using the washing procedure including 30 min incubation of the glass coverslips in the 1% aqueous Hellmanex (Hellma) solution in the ultrasonic bath, followed by the 15 min incubation in distilled water (also in the ultrasonic bath) to remove the detergent. Washed glass coverslips were kept in the distilled water and carefully rinsed and dried under pure nitrogen before using.

Silver island film was prepared on bare glass coverslips as described previously [55], utilizing wet chemistry method based on reduction in the silver nitrate (49 mM aqueous solution) with sodium hydroxide (5% aqueous solution) and D-glucose (272 mM aqueous solution) carried out in a glass beaker equipped with a Teflon-coated magnetic stirring bar and placed on a temperature-controlling plate. After the synthesis the silver-covered glass substrates were sonicated and kept in distilled water. Moreover, these substrates were carefully rinsed and dried before using.

Sample preparation was performed analogously for both substrates, bare glass coverslips and glass covered with silver island film. In this procedure the 20 µL of buffer solution of the PSI–LHCI complexes (24 µg mL^−1^) including 0.001% of the *n*-dodecyl β-D-maltoside, and 2% of the polyvinyl alcohol was spin-casted on a substrate.

### 3.3. Spectroscopic Characterization of the Materials

Absorption of PSI–LHCI buffer solution (12 µg mL^−1^) and extinction of the silver island film were carried out using Varian Cary 50 spectrophotometer. Emission spectrum of the PSI–LHCI buffer solution (1.2 µg mL^−1^) was measured with Horiba Jobin Yvon Fluorolog 3 spectrofluorimeter.

### 3.4. Fluorescence Spectroscopy

The measurements of the PSI–LHCI emission spectra kinetics were carried out using home-built fluorescence microscope. The excitation beam was provided by picosecond lasers (405 nm and 485 nm) or by supercontinuum laser source equipped with acousto-optic tunable filter (Fianium) producing laser beams at 570 nm, 580 nm, and 610 nm, all of them were additionally filtered out by FB570/10, FB580/10, FB610/40 band-pass spectral filters (Thorlabs), respectively. Moreover, in the case of all of the lines provided by the supercontinuum laser source the FES650 filter (Thorlabs) was also used. The laser beam was reflected by the dichroic mirror (650LP) and directed to the LMPlan FLN 50x objective lens (Olympus, NA = 0.5) used for excitation and collection of emitted light. The sample was mounted in front of the objective lens on a XYZ stage. In the case of the measurements for samples placed in a vacuum, there was additionally utilized a home-built vacuum chamber assembled on the XYZ stage, providing the atmospheric pressure lowered to 1.4 × 10^−8^ bar. The emission collected by the objective lens passed through the dichroic mirror and the long-pass filter, HQ655LP (Chroma), to the Amici prism used as dispersive element, and the iDus DV 420A-BV CCD camera (Andor) was utilized as the detector.

## 4. Conclusions

We reported the studies on the spectral dependence of the photostability of PSI–LHCI complexes and on the impact the plasmonically active substrate and oxygen presence have on them. We showed that there is strong connection between the wavelength of the light used for illumination and the rate of the photosystems photodamage in all of the investigated conditions. The best photostability was observed for the irradiation corresponding to minute absorption of the biomolecules, suggesting not thermal, but photoinduced degradation of the photochemical functions of photosystems. We showed that fluorescence intensity of the PSI–LHCI complexes is vitally enhanced due to interaction with silver island film; however, unfortunately, it results in much faster photobleaching of the complexes (nearly total photobleaching after 300 s long irradiation). Furthermore, we presented dramatic improvement of the photostability of photosystems in oxygen-free conditions, which brings the conclusion about the dominant role of the singlet oxygen-related species on the photodamage of the PSI–LHCI. The improvement was observed for both types of substrates used, bare glass, as well as silver island film, especially on the glass on which photobleaching of around 30% of the complexes occurred during the 300 s long experiments. Finally, we showed that the benefits arising from utilization of the plasmonically active substrate, resulting in stronger absorption of the PSI–LHCI complexes (and thus more intense emission thereof), overcome the effect of faster photobleaching in such a case, and even over a 10-fold increase in the total photons emitted by PSI–LHCI can be observed due to the interaction thereof with silver island in oxygen-free conditions. Our results show the promising ways toward optimization in the design of developed bioinspired solar fuels, taking full advantage of the application of plasmonically active materials, and limiting the destructive impact of photobleaching by providing operation in an oxygen-free environment.

## Figures and Tables

**Figure 1 ijms-23-02976-f001:**
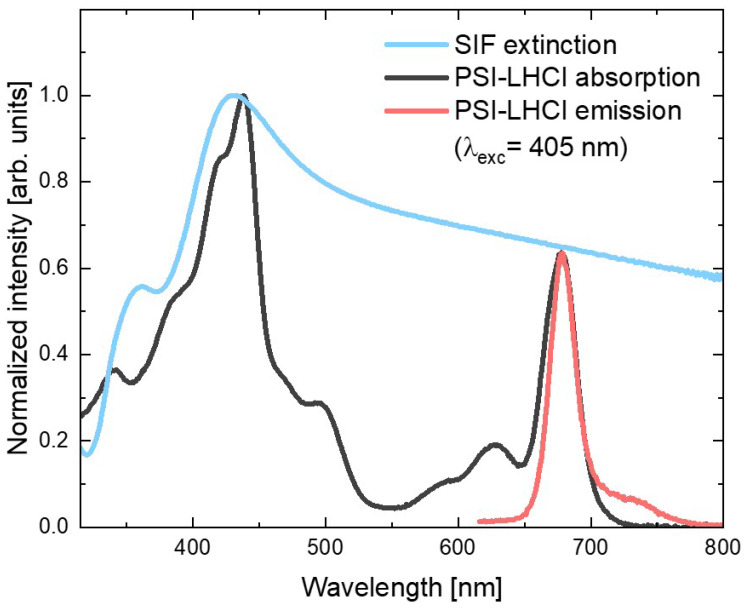
Absorption and emission spectra of PSI–LHCI (black and red, respectively) and extinction spectrum of SIF (blue).

**Figure 2 ijms-23-02976-f002:**
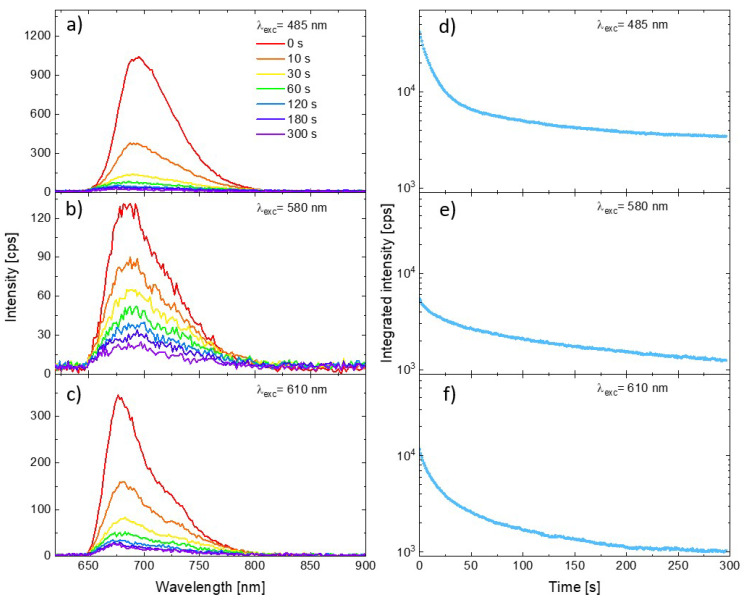
Emission spectra of PSI–LHCI complexes excited at 485 nm (**a**), 570 nm (**b**), and 610 nm (**c**) at the various moments form illumination beginning. Corresponding curves showing temporal changes in the integrated intensity of the spectra (integrated in the range from 650 to 800 nm) are shown in panels (**d**–**f**).

**Figure 3 ijms-23-02976-f003:**
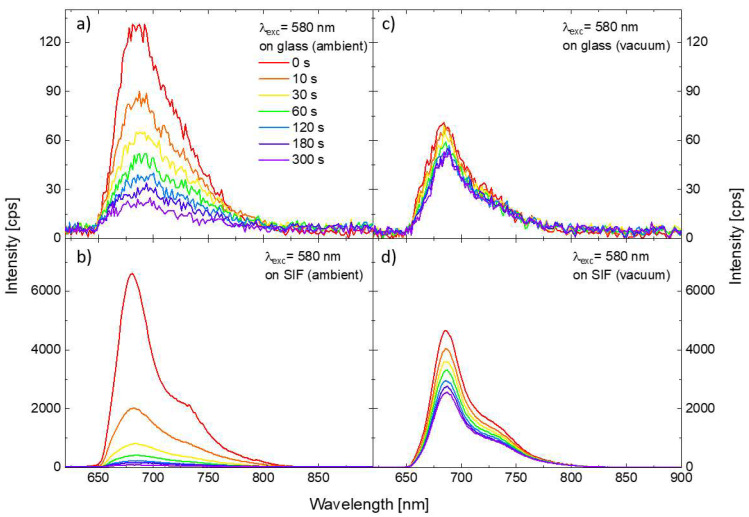
Temporal changes in the emission spectra of the PSI–LHCI layer deposited on the glass (**a**) and on the SIF (**b**) under excitation at 580 nm in the ambient conditions. Analogous results sets obtained for the samples kept in the vacuum are shown in the panels (**c**,**d**).

**Figure 4 ijms-23-02976-f004:**
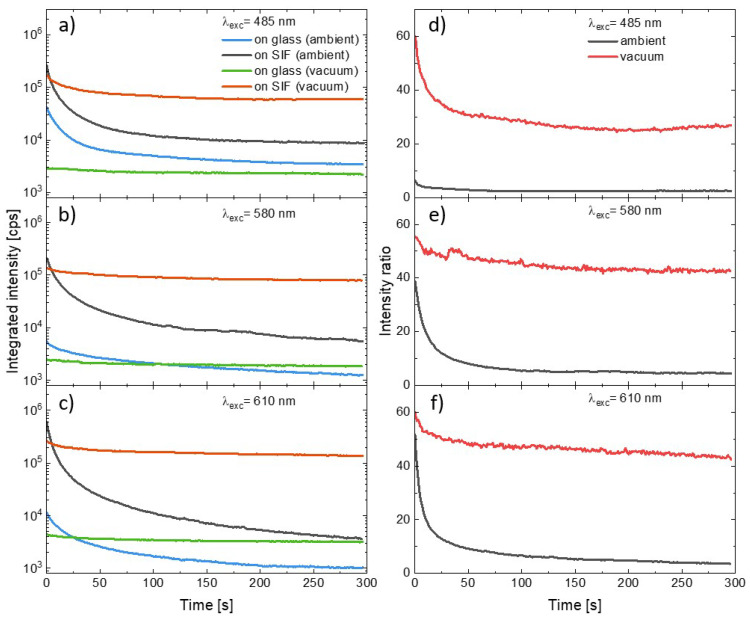
Temporal changes in the integrated intensity of the consecutive PSI–LHCI emission spectra measured for the excitation under 485 nm (**a**), 580 nm (**b**), and 610 nm (**c**). The PSI–LHCI layer was prepared on the glass (measured in ambient conditions, blue line, and in the vacuum, green line), as well as on the SIF (measured in ambient conditions, black line, and in the vacuum, red line). Corresponding temporal changes in the intensity ratio between samples prepared on the SIF and prepared on the glass are shown in the panels (**d**–**f**) (black and red lines for the measurements carried out in the ambient and vacuum conditions, respectively).

**Figure 5 ijms-23-02976-f005:**
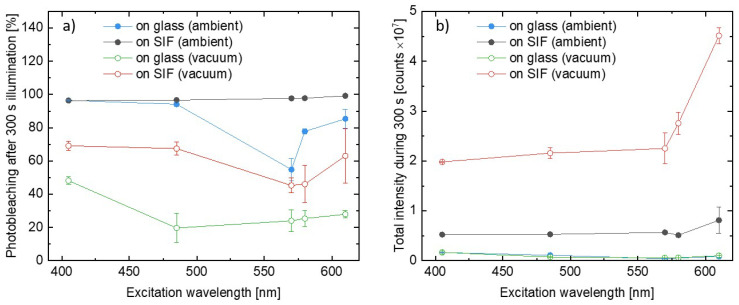
(**a**) Comparison of the relative photobleaching after 300 s of illumination in all considered conditions. (**b**) Comparison of the total light intensity emitted by the PSI–LHCI during the 300 s long illumination with laser light.

## Data Availability

Not applicable.

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
