# Peer review of "Improving Photostability of Photosystem I-Based Nanodevice by Plasmonic Interactions with Planar Silver Nanostructures"

_ijms, 2022, doi:10.3390/ijms23062976_

Round 1

Reviewer 1 Report

The following issues must be addressed:

  1. Abstract should contain some of the most representative results.
  2. It’s unusual to put the aim of this study on the page 4. The aim should be provided on the last paragraph of the Introduction part.
  3. The influence of PSI-LHCI photodamage and thermodamage should be clearly provided.
  4. Please provide more details for the following statement: “Faster photobleaching of samples prepared on the SIF also may be attributed to the plasmon-enhanced absorption and thus intensified interaction of the photosystems with light.”
  5. How you explain the significant photobleaching reduced rate when SIF substrate is involved.
  6. Explain in more details why the samples on glass placed in vacuum exhibit higher photostability.
  7. The authors use the word “Important” to often which dissipate the attentions to real important findings. I suggest making some corrections in this regard.

Author Response

We thank the Reviewer for a positive evaluation of our work. We have carefully considered all of the comments and amended the manuscript accordingly. We believe that after the revision the manuscript is suitable for publication.The following issues must be addressed:

  1. Abstract should contain some of the most representative results.

Response: Thank you for this comment, additional numerical and quantitative information, as well as the application context of this research are all included in the revised version of the abstract.

  1. It’s unusual to put the aim of this study on the page 4. The aim should be provided on the last paragraph of the Introduction part.

Response: Thank you for this comment, the information about the aim of the research is included in the revised version of the abstract.

  1. The influence of PSI-LHCI photodamage and thermodamage should be clearly provided.

Response: In the manuscript we discuss that the dominant mechanism responsible for observed degradation of the photosynthetic proteins is photobleaching due to interaction of the excited state molecules with oxygen. It is supported by the correlation of the photodegradation rate with absorption spectrum of the PSI-LHCI and dramatic improvement of the photostability thereof in anoxic conditions. The most probable cause of the still observed (however, substantially limited) degradation of PSI-LHCI, next to interaction with the small number of oxygen molecules still present in the chamber, is thermal degradation due to heat produced by the focused laser spot. Thermal denaturation of the proteins should be expected in the temperatures above 60°C, typically, therefore it should be considered to be one of the main factors responsible for degradation of the PSI-LHCI and not related directly to the presence of oxygen. It was not analysed in details in current work, which was focused on the effects connected to photobleaching of the molecules.   

  1. Please provide more details for the following statement: “Faster photobleaching of samples prepared on the SIF also may be attributed to the plasmon-enhanced absorption and thus intensified interaction of the photosystems with light.”

Response: Broader description of this statement was provided in the revised version of the manuscript. Deposition of the PSI-LHCI complexes on the plasmonically active planar metallic nanostructure (SIF) is expected to result in plasmon-induced modification of the optical properties of the PSI-LHCI. There can be in particular expected to observe in such a system plasmon enhancement of the PSI-LHCI fluorescence intensity, which in turn may arise from either enhanced emission rate or from enhanced absorption rate of the emitters. Here the latter is expected to have the dominant contribution, as reported for such structures previously, and as might be expected from spectral relations between the extinction spectrum of the SIF and absorption and emission spectra of the PSI-LHCI. The mechanism related to that is local plasmon-induced strong enhancement of the local electric field. Thus, due to such field enhancement more intense excitation of the PSI-LHCI complexes occurs, however, it also gains the photodamage of the molecules. Enhanced local field intensity is expected to bring analogous results as corresponding increase of the excitation light power.

  1. How you explain the significant photobleaching reduced rate when SIF substrate is involved.

Response: We observed increased photobleaching of the PSI-LHCI complexes when they were deposited on the SIF substrate in comparison to corresponding reference structures without SIF.

  1. Explain in more details why the samples on glass placed in vacuum exhibit higher photostability.

Response: We believe that this aspect of interpretation of experimental results is adequately described in the manuscript.

  1. The authors use the word “Important” to often which dissipate the attentions to real important findings. I suggest making some corrections in this regard.

Response: We exchanged the wording in the revised version of the manuscript.

Reviewer 2 Report

The manuscript deals with an attractive research topic, having a sound methodological analysis and revealing important findings in the field of manufacture designing solar cells and solar fuel cells under more efficient solar energy conversion. To this end, the manuscript can be accepted for publication at the “International Journal of Molecular Sciences” after the consideration of the following review comments.

  • The Abstract section can be accompanied by numerical data or quantitative information that has been derived from the conducted analysis. Besides, the statement of “application of the biomolecules in such a context is strongly limited by the progressing photobleaching thereof during illumination. In the current work we investigated the excitation wavelength dependence of the photosystem I photodamage dynamics, including the scenario with the solar energy converting performance enhanced due to utilization of the plasmonically-active substrate. Finally, we present a rational approach for the significant improvement of photostability of PSI in anoxic conditions”, it can be accompanied by signifying what should be the technological or manufacturing applications of the conducted analysis at “designing efficient and stable bioinspired solar cells” into real world conditions? Two or three concluding sentences at the Abstract section are adequate.
  • The Introduction section is detailed and well organized. However, the narrative is deprived from numerical data or quantitative information that should better support the relevant statements. Therefore, authors are recommended to convey their theoretical coverage with selected numerical-performance-operational data, either next to the already presented plenary text, or in the form of an aggregated Table at the end of the Introduction section.
  • At the end of four subsections of the main section 2 there should be a collective Table in which all reagents involved in analysis, along with that of microalga C. merolae and that of PSI-LHCI complexes formed in a polymer matrix, to be collectively represented in the form of molecular types and the active chemical bonds of functioning.
  • At subsection 2.1 the statement of “the procedure of C. merolae culture growth, thylakoid isolation and PSI-LHCI complex purification was performed, as described in [44] using a 3-step anion-exchange chromatography” it can be developed in a more detailed-descriptive manner, enabling authors to be familiarized with the conducted analysis. Besides, the measurements and those quantitative data related to the specific analysis, they can be denoted in this extra information. To this end three or four extra sentences are adequate. The same (quantitative, procedural) enrichment of information applies also to the statement of subsection 2.2 “Silver islands film was prepared on bare glass coverslips as described previously [45], also kept in distilled water and carefully rinsed and dried before using”. Two or three extra explanatory sentences per each of the recalled citations [44] and [45], above, they can be formulated at the relevant text points.
  • The narrative of section 3. Results and Discussion it can be reorganized into shorter subheadings’ representing the main outcomes and argumentation in alignment with the:  i) photobleaching investigated, ii) PSI-LHCI photostability in vacuum achieved, iii) the PSI-LHCI photodegradation efficiency. For example, the effects of (indicatively):
  1. time of illumination of PSI-LHCI complexes and light absorbed,
  2. optical properties of red algal PSI-LHCI,
  3. PSI-LHCI spectra and SIF spectrum (in reference with): i) emission intensity and the initial spectra varied with the excitation wave length, ii) the plasmon-induced gaining of the PSI-LHCI absorption rate,
  4. specifying the exact photobleaching kinetics reported,
  5. oxygen species towards PSI-LHCI photobleaching, photosystems placed in deeper layers of the polymer matrix, and the  saturated photostability observed afterwards,
  6. oxygen-free conditions supporting dramatic improvement of the photostability,
  7. metallic nanoparticles excited with wavelengths.

The critical point here is authors to select among the aforementioned key-steps of their analysis and outcomes, and to represent them in a more systematic, subsections-structured, manner. All these topics, 1) up to 7) above, they are already addressed in section 3, thus, gathering and representation in new separated subsections into  this main section 3 it is suggested. In case authors wish to creatively expand their narrative per thematic selected, they could expand the existing text content of section 3 up to one extra and cross-citing text page (in the form of subsections, accordingly). Besides, authors are recommended to provide the constraints, the limitations, the large-scale prospects and the challenging issues referred to the following groups of research objectives 1 and 2, below. To this end one extra closing subsection within main section 3 it can be added. The forecasting possibilities of adapting the research findings in real world -industrial and manufacturing- applications should be exactly determined. Three or four cross-cited sentences are adequate.

Research objectives:

  1. “.…….two competing effects related to utilization of the SIF substrates: (1) PSI-LHCI fluorescence enhancement and (2) increased PSI photobleaching rate. Our analysis of timetraces of PSI-LHCI emission intensity demonstrates that benefits from applying plasmonically-active SIF substrates surpass any negative effects resulting from intensified PSI-LHCI photodamage. This observation is highly relevant for the biosolar applications of this complex. Finally, we demonstrate the significant improvement of the PSI-LHCI photostability in anoxic (vacuum) conditions, pointing towards the crucial role of the highly reactive singlet oxygen species in inducing the photodamage of the PSI-LHCI complex”.
  2. “.……..designing solar cells or solar fuel cells, and should translate into more efficient solar energy conversion. Even in ambient conditions, the profits resulting from plasmonic interactions overcome drawbacks associated with reduced photostability. Clearly, such a balance can be significantly and positively shifted when vacuum conditions are applied  and the most destructive factor –oxygen – is eliminated”.
  • Grammar check should be taken in unifying the typing format of Figures’ legends, since some of them have been typed in italics, whereas others they have been typed in normal typing.

Author Response

The manuscript deals with an attractive research topic, having a sound methodological analysis and revealing important findings in the field of manufacture designing solar cells and solar fuel cells under more efficient solar energy conversion. To this end, the manuscript can be accepted for publication at the “International Journal of Molecular Sciences” after the consideration of the following review comments.

Response: We thank the Reviewer for such a positive evaluation of our work. We have carefully considered all of the comments and amended the manuscript accordingly. We believe that after the revision the manuscript is suitable for publication.

  • The Abstract section can be accompanied by numerical data or quantitative information that has been derived from the conducted analysis. Besides, the statement of “application of the biomolecules in such a context is strongly limited by the progressing photobleaching thereof during illumination. In the current work we investigated the excitation wavelength dependence of the photosystem I photodamage dynamics, including the scenario with the solar energy converting performance enhanced due to utilization of the plasmonically-active substrate. Finally, we present a rational approach for the significant improvement of photostability of PSI in anoxic conditions”, it can be accompanied by signifying what should be the technological or manufacturing applications of the conducted analysis at “designing efficient and stable bioinspired solar cells” into real world conditions? Two or three concluding sentences at the Abstract section are adequate.

Response: Thank you for this comment, additional numerical and quantitative information, as well as the application context of this research are both included in the revised version of the abstract.

  • The Introduction section is detailed and well organized. However, the narrative is deprived from numerical data or quantitative information that should better support the relevant statements. Therefore, authors are recommended to convey their theoretical coverage with selected numerical-performance-operational data, either next to the already presented plenary text, or in the form of an aggregated Table at the end of the Introduction section.

Response: Following the suggestion made by the Reviewer, we included additional numerical and quantitative information in the revised version of the manuscript.

  • At the end of four subsections of the main section 2 there should be a collective Table in which all reagents involved in analysis, along with that of microalga C. merolae and that of PSI-LHCI complexes formed in a polymer matrix, to be collectively represented in the form of molecular types and the active chemical bonds of functioning.

Response: We believe that all the necessary information regarding methodology of immobilization of the PSI-LHCI complex in the PVA polymer matrix on glass and glass/SIF substrates are described in detail in Materials and Methods.

4) At subsection 2.1 the statement of “the procedure of C. merolae culture growth, thylakoid isolation and PSI-LHCI complex purification was performed, as described in [44] using a 3-step anion-exchange chromatography” it can be developed in a more detailed-descriptive manner, enabling authors to be familiarized with the conducted analysis. Besides, the measurements and those quantitative data related to the specific analysis, they can be denoted in this extra information. To this end three or four extra sentences are adequate. The same (quantitative, procedural) enrichment of information applies also to the statement of subsection 2.2 “Silver islands film was prepared on bare glass coverslips as described previously [45], also kept in distilled water and carefully rinsed and dried before using”. Two or three extra explanatory sentences per each of the recalled citations [44] and [45], above, they can be formulated at the relevant text points.

Response: Description of the procedures of C. merolae growth, PSI-LHCI complex purification and silver islands film production are both presented in appropriate paragraphs in the revised version of the manuscript.

  • The narrative of section 3. Results and Discussion it can be reorganized into shorter subheadings’ representing the main outcomes and argumentation in alignment with the: i) photobleaching investigated, ii) PSI-LHCI photostability in vacuum achieved, iii) the PSI-LHCI photodegradation efficiency. For example, the effects of (indicatively):
  1. time of illumination of PSI-LHCI complexes and light absorbed,
  2. optical properties of red algal PSI-LHCI,
  3. PSI-LHCI spectra and SIF spectrum (in reference with): i) emission intensity and the initial spectra varied with the excitation wavelength, ii) the plasmon-induced gaining of the PSI-LHCI absorption rate,
  4. specifying the exact photobleaching kinetics reported,
  5. oxygen species towards PSI-LHCI photobleaching, photosystems placed in deeper layers of the polymer matrix, and the saturated photostability observed afterwards,
  6. oxygen-free conditions supporting dramatic improvement of the photostability,
  7. metallic nanoparticles excited with wavelengths.

Response: The planar text of the section 3. has been reorganized in shorter subheadings. We believe that the clarity of presentation has been substantially improved. The forecasting application possibilities of the research findings are also included in the revised version of the manuscript.

  • Grammar check should be taken in unifying the typing format of Figures’ legends, since some of them have been typed in italics, whereas others they have been typed in normal typing.

Response: Thank you for this comment, the format of all figure captions is unified in the revised version of the manuscript.

Round 2

Reviewer 1 Report

The manuscript can be published in the present form.

Reviewer 2 Report

The revised manuscript has been creatively improved, having the review comments considered in a sound and systematic manner. The: a) PSI-LHCI synthesis, characterization, photobleaching, b) the proven significant improvement of PSI-LHCI photostability in anoxic conditions, and c) the reporting negative effects resulting from intensified PSI-LHCI photodamage, they are all related to the biosolar applications of this PSI-LHCI complex. In this context, the analysis structure and the research outcomes of this revised manuscript they are revealing novel new knowledge in the field of designing and promoting efficient and stable bioinspired solar cells. Therefore, the revised manuscript can be accepted for publication at the "International Journal of Molecular Sciences" as is.